# Fabry-Perot Interferometer Based on a Fiber-Tip Fixed-Supported Bridge for Fast Glucose Concentration Measurement

**DOI:** 10.3390/bios12060391

**Published:** 2022-06-06

**Authors:** Shuo Tang, Mengqiang Zou, Cong Zhao, Yihang Jiang, Ribao Chen, Zhourui Xu, Chengbin Yang, Xiaomei Wang, Biqin Dong, Yiping Wang, Changrui Liao, Gaixia Xu

**Affiliations:** 1Guangdong Key Laboratory for Biomedical Measurements and Ultrasound Imaging, School of Biomedical Engineering Health Science Center, Shenzhen University, Shenzhen 518055, China; tangshuo2020@email.szu.edu.cn (S.T.); jiangyihang2019@email.szu.edu.cn (Y.J.); 2070246044@email.szu.edu.cn (R.C.); xuzhouray@szu.edu.cn (Z.X.); cbyang@szu.edu.cn (C.Y.); 2Shenzhen Key Laboratory of Photonic Devices and Sensing Systems for Internet of Things, Guangdong and Hong Kong Joint Research Centre for Optical Fiber Sensors, Shenzhen University, Shenzhen 518060, China; zoumengqiang2020@email.szu.edu.cn (M.Z.); zhaocong@szu.edu.cn (C.Z.); ypwang@szu.edu.cn (Y.W.); 3Key Laboratory of Optoelectronic Devices and Systems of the Ministry of Education and Guangdong Province, College of Physics and Optoelectronic Engineering, Shenzhen University, Shenzhen 518060, China; 4Base for International Science and Technology Cooperation: Carson Cancer Stem Cell Vaccines R&D Center, Shenzhen Key Laboratory of Synthetic Biology, Department of Physiology, School of Basic Medical Sciences Shenzhen University, Shenzhen 518055, China; xmwang@szu.edu.cn; 5Department of Civil Engineering, Guangdong Province Key Laboratory of Durability for Marine Civil Engineering, The Key Laboratory on Durability of Civil Engineering in Shenzhen, Shenzhen University, Shenzhen 518060, China; incise@szu.edu.cn

**Keywords:** optical fiber sensor, Fabry-Perot (FP) cavity biosensor, femtosecond laser micromachining, two-photon polymerization, blood glucose detection

## Abstract

Blood glucose concentration is important for metabolic homeostasis in humans and animals. Many diabetic patients need to detect blood glucose daily which burdens community hospitals and family healthcare. Optical fiber sensors are widely used in biomedical detection because of their compact structure, fast response, high sensitivity, low cost, and ease of operation. In this work, we constructed a Fabry-Perot (FP) cavity biosensor for the fast detection of glucose concentration in serum. The femtosecond laser micromachining was applied to fabricate the FP cavity by printing the fiber-tip fixed-supported bridge at the end face of the optical fiber. An additional hemisphere was printed at the center of the outer surface of the bridge to avoid multi-beam interference. The results demonstrated that the proposed biosensor had high refractive index (RI) detection sensitivity, roughly 1039 nm/RIU at a wavelength of 1590 nm, and the detection sensitivity for glucose was around 0.185 nm/ (mg/mL) at a wavelength of 1590 nm. Due to its high sensitivity, compact structure, and fast response, the FP cavity biosensor has great potential to be applied in family healthcare for glucose concentration detection of diabetic patients.

## 1. Introduction

It is well known that glucose is one of the most important sources of metabolic carbohydrate energy. Glucose concentration is also a useful indicator for various diseases, especially diabetes [1,2]. The global diabetes prevalence in 2019 was reported as high as 9.3% (463 million people) and would rise to 10.2% by 2030 [3,4]. As the number of diabetic patients continues to increase, the detection of blood glucose concentration has become more frequent in the community hospitals and family daily routine healthcare.

Generally, the blood glucose detection [5,6] included the electrochemical method and enzymatic method [7]. However, the sensitivity of the electrochemical method was limited by the comparatively slow kinetics of glucose electro-oxidation on conventional electrodes [8,9]. Although the enzyme-based method has improved the sensitivity and selectivity for glucose detection, the high cost and the complex immobilization process blocked the enzyme-based glucose sensors to enter families [10].

Due to their compact structure, high sensitivity, corrosion resistance, and electromagnetic immunity, optical fiber sensors have been widely investigated in the fields of food safety [11], environmental monitoring [12,13], and biomedicine [14,15]. In the past few years, various optical fiber sensors have been reported, such as fiber surface plasmon resonance (SPR) sensors [15,16] and fiber Bragg grating (FBG) sensors [17,18]. These sensors had excellent performance in refractive index RI detection of unlabeled biological samples. However, there are still several disadvantages of these widely reported sensors [19], e.g., the complex fabrication of SPR sensors, the cross-sensitivity towards temperature and RI of FBG sensors, etc. On the other hand, the optical fiber Fabry-Perot (FP) cavity sensor has the advantages of a simple detection principle, large detection range, and linear response [10,20,21], so it has wide applications in biosensing.

As reported by other research groups, fluorescent optical fiber sensors have the advantages of high specificity and high sensitivity in blood glucose detection, but their service life was short and the reaction time was long, about 1 h [22]. SPR fiber optic sensors have ultra-high detection sensitivity, but their fabrication and calibration are complicated [23]. Compared with other fiber optic sensors, our device responded fast and had a considerable sensitivity for glucose detection [24,25]. What is more, such a sensor did not undergo complex surface modification and specific labelling, which simplifies the fabrication of the device and improves reproducibility.

Two-photon printing (TPP) technology induced by a femtosecond laser is a new type of 3D microfabrication technology [26,27]. It has been applied in micro-instruments [28,29], microfluidics [11,30], etc. Due to the ultra-high printing accuracy of TPP technology (typical resolution is less than 100 nm), some functional microstructures that were difficult to fabricate by using conventional micromachining technology had been printed, such as micro-biomimetic or micro-magnetic driving mechanical structures [31,32,33]. By applying TPP technology, various new types of optical fiber sensors [21] have also been developed and reported in recent years. For example, Melissinaki et al. fabricated a micro FP cavity on the end face of an optical fiber as a vapor sensor [34]. Xiong et al. applied TPP to make a microcantilever for hydrogen detection [35]. Zou et al. printed a clamped-beam on the end face of the optical fiber for tiny-force measurements [36]. However, the above-mentioned optical fiber sensors were all implemented in a gas environment, while biomedical detections were mostly applied for liquid samples. Therefore, achieving stable and accurate detection in liquid environments was necessary for the development of biomedical sensors.

In this work, we proposed a novel “fiber laboratory” on the end-face of optical fiber by applying TPP technology for glucose detection (Figure 1). The polymer fixed-support bridge was printed on the end face of the single-mode fiber (SMF). A simple FP cavity was formed and it was sensitive to the refractive index changes of analytes. Then, the concentration of glucose in the solution was measured by the FP interferometer based on the fiber-tip fixed-supported bridge. The results demonstrated that the proposed compact sensor responded fast to the refractive index changes and had high sensitivity to the glucose concentrations, which provided a good alternative for easy-use glucose detection in both community hospitals and family routine healthcare.

## 2. Materials and Methods

### 2.1. Fabrication

The photoresist used in this work was prepared as in [37]: IGR-369 (Ciba-Geigy, Basel, Switzerland) was used as the photoinitiator. The trifunctional monomers SR444, SR368, and SR454 (from Sartomer, Exton, PE, USA) were mixed and dissolved in acetone. 4-hydroxyanisole (purchased from Sigma-Aldrich, St. Louis, MI, USA) was used as a polymerization inhibitor. Tetraethyl thiuram disulfide, purchased from Sigma-Aldrich, was used as an accelerator. The polymer material employed has a refractive index of ~1.53 [35].

To fabricate the fixed-supported bridge, an SMF (Corning SMF-28, 8.2 μm core diameter) was fixed on a glass slide and its tip was immersed into the photoresist, as shown in Figure 2a. The polymerization process of the micro FP cavity was adopted and shown in Figure 2b [35,36]. A two-photon polymerization 3D lithography machine (Altechna R&D WOP, FemtoLAB) was used to fabricate structures. A femtosecond laser with a pulse width of 250 fs, a center wavelength of 1026 nm, and a repetition frequency of 200 kHz was applied to perform TPP. A 63× magnification (NA = 1.4) oil-immersion objective lens was used to increase the fabrication accuracy and the smoothness of the microstructure. In this work, the real light spot size was about 0.5 μm and the focal depth was about 3 μm. To ensure the strength of the fabricated microstructure and minimize time cost, the laser power was set at 2 mW and the scanning speed was set at 500 μm/s, respectively. After the polymerization was completed, the microstructure was washed by washing solution (acetone: isopropanol, 1:4) to remove the residual photoresist (Figure 2c). After all the washing solution was evaporated, the microfabrication of the FP cavity was completed.

### 2.2. Optimization and Characterization

To achieve the best glucose sensing performance, a series of fiber-tip fixed-supported micro-bridge with different thicknesses and heights were fabricated and tested. The thickness of the fixed-supported micro-bridge was chosen to be 10 μm since the thinner bridge would easily bend due to the tension induced by the solution evaporation when changing the test samples. What is more, it was confirmed that the 10 μm thick micro-bridge did not deform after all the solution on it has evaporated by optical microscope and spectrum shift (Appendix A).

Since the free spectral range (*FSR*) of interference reflection can be applied to calculate the effective cavity length of the FP cavity sensor, we measured the *FSR* of the fiber-tip FP interferometer (Figure 3). The reflection spectra of various FP interferometers showed that the *FSR*s were 50.18 nm, 25.38 nm, and 15.04 nm according to the different dip wavelength λ of 1383 nm, 1375 nm, and 1324.5 nm, respectively. There was a relationship with these parameters according to the formula:(1)$$FSR=\frac{\lambda^{2}}{2nL}$$
where *λ* is the dip wavelength, n represents the average RI within the cavity, and *L* is the cavity length. Thus, the effective cavity length of the FP cavity sensor (*L*) was calculated to be 58.23, 37.25, and 19.06 μm, corresponding to the designed cavity length of 60, 40, and 20 μm, respectively. The machining precision was acceptable in this work. It was obvious that the shorter the cavity length was, the more light was reflected back to the fiber to participate in the interference, which will reduce the light loss and make the contrast higher. Herein, the FP interferometer with a cavity length of 20 μm had the best spectrum contrast. However, in practical terms, if the cavity length was small, e.g., 20 μm, it was difficult to wash the cavity or put the sample solution completely into the cavity. Thus, we optimized the parameters of the sensor as the following, FP cavity with a length of 35 μm, and a fixed-support micro-bridge with a length of 110 μm, a width of 20 μm, and a thickness of 10 μm.

The scanning electron microscope (SEM) images provided the details of the fixed-supported bridge (Figure 4). It was clear that the fixed-supported bridge structure was firmly printed on the flattened fiber end face. The contact surface between the fiber end-face and the air was defined as the reflective surface 1, and the contact surface between the air and the inner surface of the micro-bridge was defined as the reflective surface 2. Reflecting surfaces 1 and 2 formed a simple FP cavity. Moreover, to avoid multi-beam interference caused by the reflection of the light from the third surface (the outer surface of the beam), an additional hemisphere was polymerized at the center of the third surface. This hemisphere could reflect the light reaching the outer surface to the outside of the fiber core and eliminate the unnecessary interference (Appendix A). Such design simplified the FP cavity as a double-beam optical interference. The reflected light intensity can be calculated as [38]:(2)$$I=I_{1}+I_{2}+2\sqrt{\left(I_{1}+I_{2}\right)}cos\left(\frac{2\pi Ln}{\lambda }+\Phi_{0}\right)$$
where *I*_1_ and *I*_2_ are the reflected intensities from surfaces 1 and 2, and Φ_0_ is the initial phase of interference. According to Equation (2), it can be concluded that when the cosine term is equal to an odd multiple of π, the reflected light intensity reaches the minimum value, namely:(3)$$\mathrm{when}\;\frac{2\pi Ln}{\lambda }+\Phi_{0}=\left(2m+1\right)\pi ,\;I=I_{min}=I_{1}+I_{2}-2\sqrt{\left(I_{1}+I_{2}\right)}$$
where *m* is an integer. When the cavity length is a constant, the corresponding relationship between the shift of the dip wavelength and the change of the average RI in the FP cavity is as follows:(4)$$\frac{d\lambda }{dn}=\frac{\Delta \lambda }{\Delta n}$$

Here, Δ*n* is the effective RI variation of the medium in the microcavity and Δ*λ* is the amount of shift in dip wavelength due to changes in the effective refractive index. However, due to the focal depth problem of two-photon laser polymerization, the designed polymer hemisphere was stretched along the *z*-axis and fabricated as a half ellipsoid. Fortunately, such distortion had no adverse effect on our device and the desired double beam interference could be obtained. The SEM images confirmed the smooth mess of the micro-bridge surface and the excellent parallelism between the fiber end-face and the bridge surface (Figure 4), which ensured both strong reflected light intensity and contrast of the device.

## 3. Results

### 3.1. RI Response Measurement

Before measuring blood glucose concentration, the sensor was calibrated using standard RI liquids (Cargille Labs, Cedar Grove, NJ, USA). The experiments were carried out in a cleanroom where temperature and humidity were stable. The fiber-tip was moved into a droplet of standard RI liquid on a clean glass slide by a three-axis displacement platform. The entire testing was monitored using optical microscopy to confirm the FP cavity was filled by the measured solution. The ethanol solution was used to remove the remaining RI liquid. To cover the RI range of the typical liquid biological sample, the RI of the standard liquids ranged from 1.32 to 1.35, with an interval step of 0.005.

Figure 5a showed the reflectance spectrum when the RI was increased at room temperature. As the RI was increased, the spectrum dip wavelength drifted steadily toward the long wavelength. At 1590 nm, when the RI changes from 1.32 to 1.35, the dip wavelength increased about 32 nm. The dip wavelength shift was plotted as a function of RI in Figure 5b, in which the curve was well fitted by a linear fitting (R^2^ = 0.9974). The RI sensitivity of the sensor was as high as 1039 nm/RIU.

The detection limit (*DL*) for the refractive index can be calculated using the theory proposed by I. M. White and X. Fan [39]:(5)$$DL=\frac{R}{S}$$
where the *S* is the sensitivity of the sensor, and *R* is the sensor’s resolution. Typically, there were some factors that affect the sensor’s resolution including OSA resolution, the signal-to-noise ratio, and the width of the resonance. The numerical results can be approximated by *R* = 3 σ, where the standard deviation of the resulting spectral variation σ:(6)$$\sigma =\frac{\Delta \lambda }{4.5\left(SNR^{0.25}\right)}$$
where Δ*λ* is the full-width at half-maximum (FWHM) of the fringe and SNR is the signal-noise ratio. *SNR* is in linear units in this expression (e.g., 60 dB = 10^6^). In our refractive index sensing experiments, the measured value of FWHM was 6.9 nm. Assuming an SNR of 50 dB, the detection limit of this proposed sensor was 2.4 × 10^−4^ RIU, which was sensitive enough for most blood samples [8].

### 3.2. Blood Glucose Concentration Response Measurement

The typical samples for measuring the blood glucose concentration include whole blood, serum, and plasma. In this paper, we used serum as the test sample. A series of glucose solutions were prepared by dissolving glucose in fetal bovine serum (FBS, Gibco).

As shown in Figure 6, the sensor-tip was dipped in the glucose solution by a three-dimensional displacement platform. Figure 6a shows a glucose molecular diagram. Figure 6b shows our test schematic, a broadband light source was used as the transmitter and an optical spectrum analyzer was used to record the reflection spectrum. Figure 6c shows the reflection spectrum drift as a function of the measured glucose concentration. Once the FP cavity was filled with the glucose solution, the reflection spectrum drift was immediately recorded and the glucose concentration was calculated. Figure 7a shows the reflection spectrum as a function of glucose concentration. Around the wavelength of 1589 nm, when the glucose concentration changed from 1 mg/mL to 20 mg/mL, the reflection spectrum shifted to the long-wavelength by approximately 3.5 nm. Figure 7b shows the linear curve fitting of the dip wavelength shift as a function of glucose concentration, and it can be obtained that the glucose detection sensitivity of the sensor was 0.18473 nm/(mg/mL) (R^2^ = 0.9998). It can be calculated that the detection limit of the sensor for blood glucose was 0.4565 mg/mL, which was sufficient for glucose concentration detection of diabetic patients in family healthcare [8].

### 3.3. Temperature Measurement

In order to guarantee the stability of the sensor in daily monitoring, we analyzed the temperature response of the sensor. The sensor was placed into an air oven and we gradually increased the temperature from 25 °C to 65 °C, and the temperature was continuously maintained for 10 min in each step. The change in reflectance spectrum with temperature is shown in Figure 8a. The corresponding relationship between the dip wavelength shift of the Fabry-Perot interference spectrum and the temperature is as follows:(7)$$\frac{d\lambda }{dT}=\frac{2}{k}\left(\frac{dn}{dT}L+\frac{dL}{dT}n\right)$$
where *k* is an integer representing the order of the interference spectrum and *dn/dT* and *dL/dT* are the thermo-optic coefficient and thermal expansion coefficient of the medium in the cavity, respectively. When the ambient temperature increases, the interference spectrum will shift due to the two main influencing factors, the thermo-optic effect and the thermal expansion effect. In this work, our temperature tests were performed in an air medium, where temperature variations had less effect on the refractive index of the air. Thus, we deduce that the cavity length changes were due to the thermal expansion of the fixed-supported bridge in this work. During the glucose detection, the experiments were performed in an ultra-clean studio with constant temperature and humidity. In a real house or hospital environment, the temperature would not change drastically when the test would be performed.

As shown in Figure 8, our sensor exhibited very good detection stability around room temperature. According to the structure of the FP cavity, we speculated that the inward expansion of the bridge was compensated by the outward expansion of the base, in which there was no change in the FP cavity length, from 25 °C to 40 °C. With the increase in temperature, the base expanded outward more than the bridge did inward and the cavity length increased. Thus, the dip wavelength of the spectrum showed a significant red shift. Figure 8b illustrated the linear fit of the dip wavelength versus temperature (35 °C to 65 °C), achieving a calibrated drift of ~260 pm/°C (R^2^ = 0.985). Since the sensor would be applied in community hospitals and family healthcare at room temperature, the small fluctuation of temperature response could be ignored under 35 °C. If the sensor is applied above 35 °C, the calibrated drift could be used to calibrate the data.

## 4. Conclusions

In this work, we fabricated a novel FP cavity optical fiber sensor for fast glucose concentration detection by using TPP to print a fixed-supported bridge on the fiber end face. The result demonstrated that the RI sensitivity of the proposed sensor was as high as 1039 nm/RIU. The detection sensitivity of glucose solution was 0.185 nm/(mg/mL), and the concentration detection resolution was 0.4565 mg/mL, which is sufficient for glucose concentration detection of diabetic patients. Thus, such an FP cavity sensor has great potential to be applied to community hospitals and family healthcare.

## Figures and Tables

**Figure 1 biosensors-12-00391-f001:**
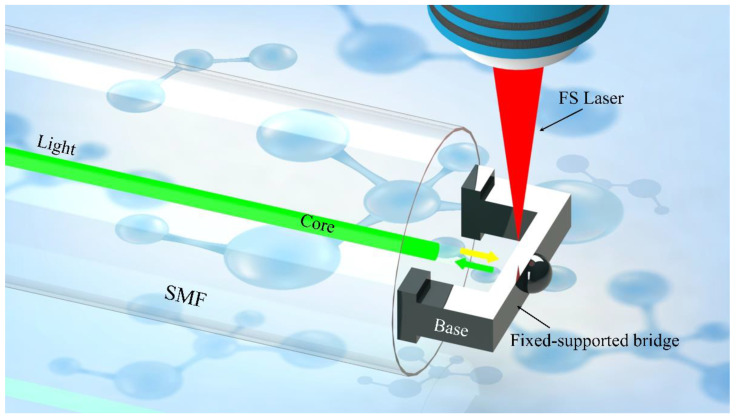
The schematic design of the all-fiber glucose sensor is based on the FP cavity.

**Figure 2 biosensors-12-00391-f002:**
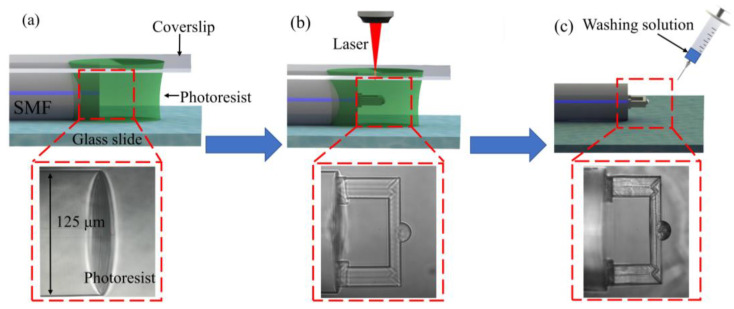
Process flow of fabricating the fiber-tip fixed-supported micro-bridge. (**a**) The fiber end face was cut to flat and immersed in the photoresist. A cover glass was set above the photoresist. (**b**) The fixed-supported bridge was printed on the end face of the optical fiber using TPP technology. (**c**) The cleaning solution was applied to wash off the remaining photoresist.

**Figure 3 biosensors-12-00391-f003:**
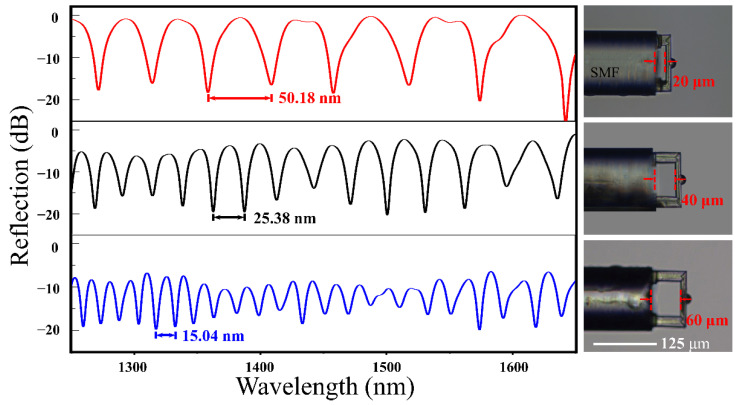
The typical reflection spectra of the FP interferometer and the optical microscopic images of the FP cavity with various cavity lengths. Scale bar: 125 μm.

**Figure 4 biosensors-12-00391-f004:**
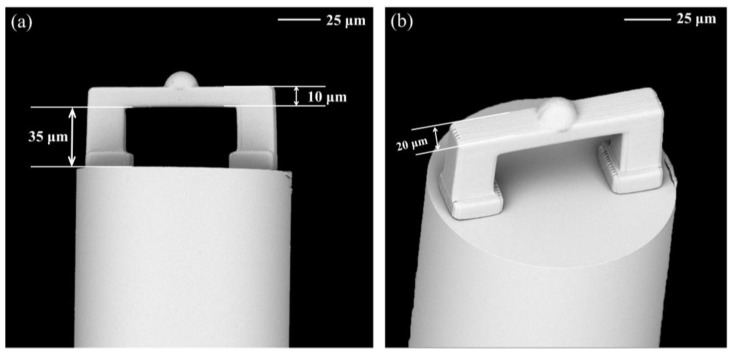
Typical SEM images of the microstructure on the end of the optical fiber manufactured using TPP technology. (**a**) Front view and (**b**) top view of the fixed-supported bridge. Scale bar: 25 μm.

**Figure 5 biosensors-12-00391-f005:**
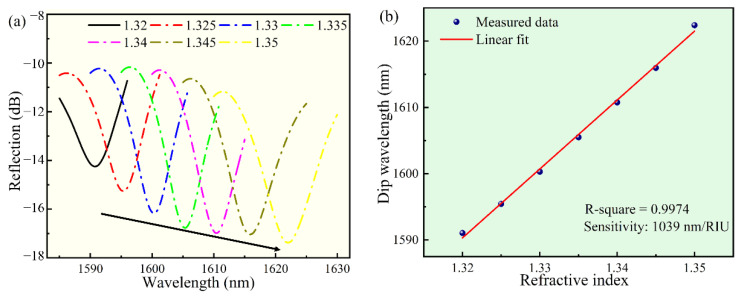
The response of the sensor to various RI solutions, in the wavelength range of 1585--1630 nm. (**a**) The dip wavelength shift of the reflectance spectrum of the sensor. (**b**) The corresponding relationship between the dip wavelength and the RI.

**Figure 6 biosensors-12-00391-f006:**
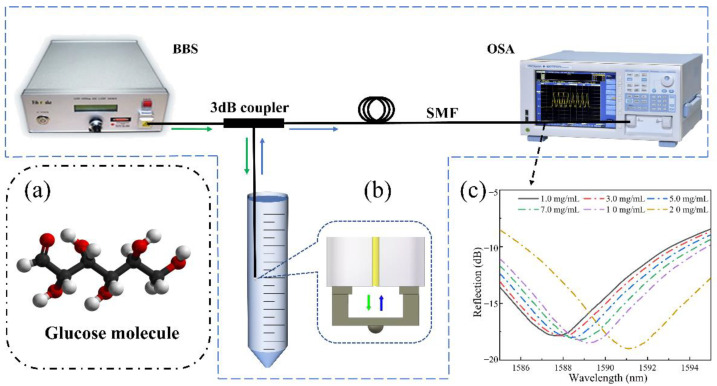
The sensing system is based on a fiber end-face fixed-supported bridge for measuring the glucose concentration: (**a**) A glucose molecular diagram. (**b**) A schematic diagram of the glucose solution detection system. (**c**) Reflection spectrum drift as a function of the measured glucose concentration.

**Figure 7 biosensors-12-00391-f007:**
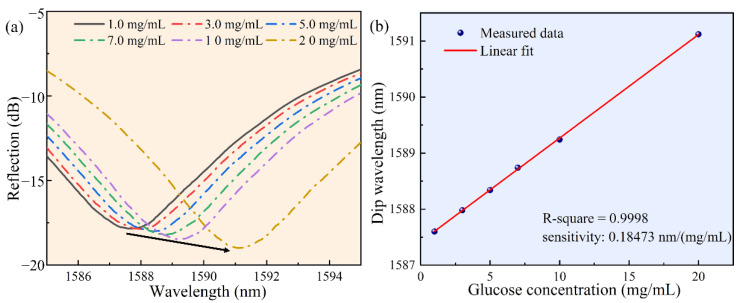
The results of measuring the glucose concentration in serum solution, under the wavelength ranging from 1580 to 1600 nm. (**a**) The reflectance spectrum as a function of glucose concentration. (**b**) The linear curve fitting of the dip wavelength and glucose concentration in serum.

**Figure 8 biosensors-12-00391-f008:**
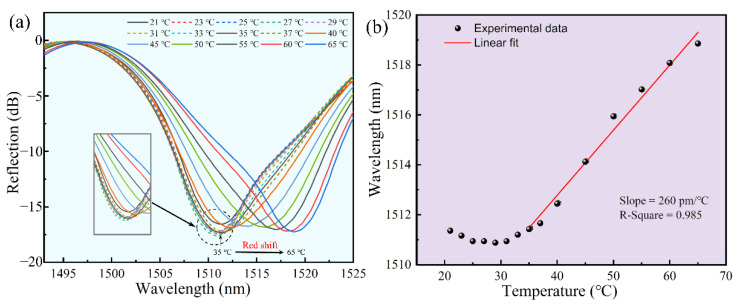
Temperature response of the polymer fixed-supported bridge (~1510 nm). (**a**) Reflection spectrum evolution of the polymer fixed-supported bridge while the temperature increases from 25 °C to 55 °C. (**b**) Data and a linear fit of the dip wavelength versus temperature from 35 °C to 65 °C.

## Data Availability

The experimental data is contained within the article.

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
