# Peer review of "Fabry-Perot Interferometer Based on a Fiber-Tip Fixed-Supported Bridge for Fast Glucose Concentration Measurement"

_biosensors, 2022, doi:10.3390/bios12060391_

Round 1

Reviewer 2 Report

Detection of blood glucose concentration is important in community hospitals and family healthcare. This manuscript reported a novel glucose sensor based on optical fiber. The femtosecond laser micromachining was applied to fabricate the FP cavity by printing the fiber-tip fixed-supported beam at the end face of the optical fiber. An ingenious hemisphere was printed at the center of the outer surface of the beam to avoid multi-beam interference. The results showed that such biosensor had a high refractive index (RI) detection sensitivity and a considerable detection sensitivity for glucose, which had great potential to be applied in family healthcare for glucose concentration detection.

However, the manuscript can be accepted after the questions and concerns below are addressed.

  1. There are some writing typos in the manuscript, and I point out the following: please check and correct carefully.

Page 3, line 88: “The photoresist used in this work was and prepared”

Page 4, line 125:In Formula 1, the “L” should be capitalized

Page 5, line 153:In Formula 2, “Φ0” should be changed as “Φ0

  1. The recent reports of other groups about the fiber optic sensors for detecting glucose concentration should be mentioned in the introduction.
  2. The actual size of the sensor should be marked accordingly in Figure 4.
  3. Figure 6 should be mentioned in manuscript.
  4. Please introduce the refractive index of the polymeric materials in the manuscript, and analyzed whether it affect the results. Why?
  5. What is the key parameter affecting the performance of your proposed biosensor?
  6. During the fabrication process, how does the base adhere to the SMF end face?
  7. You can refer to some recently published high-level articles, such as Non-Invasive, and Ultrafast Radio Frequency Biosensor Based on Optimized Integrated Passive Device Fabrication Process for Quantitative Detection of Glucose Levels. Sensors, 2020, 20(6):1565; Permittivity-Inspired Microwave Resonator-Based Biosensor Based on Integrated Passive Device Technology for Glucose Identification. Biosensors 11 (12), 508.

Reviewer 3 Report

This work describes the FP cavity by printing the fiber-tip fixed-supported beam at the end face of the optical fiber to test glucose concentration. This paper describes the manufacturing process in detail and the refractive index and temperature response are investigated. There are some points and notes the authors should answer as pointed below.

  1. Pay attention to tenses in the manuscripts. Keeping your writing in the same tense.
  2. In “Abstract”, Page 1, line 29-30: “...roughly 1039 nm/RIU at the wavelength of 1550 nm...was around 0.185 nm/ (mg/mL) at the wavelength of 1590 nm.” Why are there two detecting wavelengths?
  3. In “Introduction”, Page 2, line 66-70: “For example, ...”Do you have examples of the foreign researches?
  4. In “Introduction”, the research development of optical fiber sensor for detecting glucose concentration isn’t described. I suggest that this section is added.
  5. Page 3, line 88: “The photoresist used in this work was and prepared”and line 93:“an SEF...” These are writing mistakes. Please revise.
  6. Page 4, line 116-117: How to verify this statement “What s more, it was confirmed that the 10 μm-thick micro-beam did not deform after all the solution on it has evaporated by optical microscope and spectrum shift.”?
  7. Page 4, line 125: In Formula 1, the “L”should be capitalized.
  8. All formulas require the formula editor to edit.
  9. Page 4, line 135-136: “FP cavity with length of ...”and “fixed-support micro-beam with a length of...”What are these two lengths? Please mark the length parameter on the Figure.
  10. Page 6, line 204-205: How to verify this statement “In our refractive index sensing experiments, the measured value of FWHM was 6.9 nm.”?
  11. Figure 6a-c is not introduced in the manuscript.
  12. How to prove the fast detection of glucose concentration

Reviewer 4 Report

This manuscript proposed a Fabry-Perot (FP) cavity biosensor for fast detection of the glucose concentration in the serum. TPP was used to fabricate the fiber-tip fixed-supported beam at the end face of the optical fiber. Experimental results were presented. However, there are still some problems with this article. The specific comments are as follows:

  1. Line 129, the unit should be mm instead of nm。
  2. Line 128-129,“corresponding to the designed cavity length of 60, 40 and 20 μm, respectively.” It should be “corresponding to the designed cavity length of 20, 40 and 60 μm, respectively.”And these values of 58.23, 37.25, and 19.06 can not be calculated using Equation (1). Please check.
  3. Line 135-137,all these structural parameters should be represented by a letter and marked in Figure. 3
  4. Figure.6c and Figure.7a are both reflection spectrum when measuring the glucose concentration in serum solution. Why are they different?
  5. Line 219, “Once the FP cavity was filled with the glucose solution, the reflection spectrum drift was immediately recorded and the glucose concentration was calculated.” In my opinion, it takes time for the liquid to fully fill the FP cavity, and it is important to ensure that the liquid is fully filled into the FP cavity. Therefore, microscopy pictures are required and the spectrum should be monitored for a few minutes to make sure it is stable.
  6. Please check Equation (7) and add some derivation process appropriately.
  7. Please analyze the reason why the effect of temperature on the wavelength shift in Fig. 8b is significantly larger in the high temperature part than in the low temperature part.
  8. What is the material for the fixed-supported beam?

Round 2

Reviewer 1 Report

The authors diligently prepared answers to various questions asked by reviewers. It is judged that the technical problems and limitations in the application were clearly explained, and the directions for improvement of the developed sensor problems were sufficiently presented by appropriately suggesting alternatives. We thank the authors for their efforts to edit the manuscript.

Reviewer 4 Report

The authors have answered all the questions and revised the manuscript accordingly.